# Patterns and predictors of outcome monitoring amongst link workers: Learnings from the National Social Prescribing Link Worker Survey 2025

**Daisy Fancourt**[iD]*, **Feifei Bu**

Department of Behavioural Science and Health, University College London, London, United Kingdom

* d.fancourt@ucl.ac.uk

## Abstract

### Background

Effective outcome monitoring by Social Prescribing Link Workers (SPLWs) is critical for understanding the impact of social prescribing (SP) interventions. Despite national guidance mandating outcome reporting via the Social Prescribing Information Standard, it is unclear how consistently outcomes are measured across services in England.

### Methods

This analysis used data from the 2025 National Social Prescribing Link Worker Survey, comprising responses from SPLWs (N = 409) who hold a caseload and are based in England. This analysis focused specifically on the questions pertaining to impact monitoring and its predictors. Descriptive analyses and both multiple logistic and original regression analyses sequentially adjusted for confounders were used to explore patterns of outcome monitoring, predictors of reporting behaviour, and potential solutions to enhance data collection.

### Results

Just 57.5% of SPLWs reported recording or monitoring patient-level outcomes "often" or "very often," and 45.5% did so for service-level outcomes. Only 48.7% reported using quantitative outcome measures such as ONS4, while 91% captured qualitative data. Among those using outcome measures, 61% used ONS4, with additional use of MyCaW, PAM, and Wellbeing Star. Outcomes were typically recorded at the beginning and end of patient interactions (60.2%), but 20.4% collected data at every session.

Outcome reporting was not significantly predicted by education, training, job seniority, or most demographic variables. However, those aware of the SP Information

**Data availability statement:** The National Social Prescribing Link Worker Survey dataset is held by the National Academy for Social Prescribing and contains de-identified participant data. However, due to the relatively small sample size and potential for identification through small cell sizes across demographic and contextual variables, public sharing of the full dataset is not possible under current data governance and ethical safeguards. A fully anonymized version of the dataset is available to researchers upon reasonable request. Requests should be submitted in writing to the National Academy for Social Prescribing at: evidence@nasp.info.

**Funding:** This analysis was supported by funding from the ESRC-funded SPIRITED project [UKRI1717] and the National Academy for Social Prescribing. The funders had no role in study design, data collection and analysis, decision to publish, or preparation of the manuscript.

**Competing interests:** The National Academy for Social Prescribing provided funding for the analysis of the data, but had no say in the design, conduct or write-up of the statistical findings.

Standard were 2.3 times more likely to report outcomes (OR = 2.29, 95% CI [1.53, 3.42]), and those who shared outcomes with managers or other stakeholders were over six times more likely to monitor impact (OR = 6.03, 95% CI [2.93, 12.43]). SPLWs who reported outcomes being used to inform investment decisions were 1.8 times more likely to monitor outcomes (OR = 1.76, 95% CI [1.03, 3.02]). Notably, those with access to patient records were significantly less likely to use outcome measures (OR = 0.22, 95% CI [0.11, 0.41]).

## Conclusions

Despite modest improvements since 2023, SPLWs' monitoring of SP outcomes remains inconsistent and below national standards recommended in the SP Information Standard. There was little differentiation in reporting patterns by demographics or job role, but findings suggest increasing awareness of the SP Information Standard and simplifying reporting processes may support better data quality. These findings provide benchmarks to inform future policy and practice aimed at standardising SP outcome reporting nationally.

## Introduction

Social prescribing (SP) and community-based support is part of the NHS Long Term Plan's commitment to embed universal personalised care across the health and care system [1,2]. In 2023, NHS England commissioned the Professional Record Standards Body (PRSB) to develop an information standard to enable SP support provision to be recorded, amended and maintained collaboratively between patients, link workers and health and care professionals. The first version of the information standard mandated the use of the PRSB SP standard (version 1.0) in appropriate care settings, to ensure consistent information capture for SP activities. Part of the standard is a minimum dataset, which mandates that data is captured on patient demographics, needs and concerns, support offered (including referred to activities), and outcomes. This requirement to monitor outcomes has been reiterated in subsequent documents, such as NHS England's reference guide for primary care networks, giving the responsibility specifically to link workers (LWs) [3].

The gathering of administrative data on SP is vitally important to tracking the scale, reach and impact of the service. In the past two years, we have seen the first analyses of SP that make use of large-scale administrative data, including electronic patient health records (e.g., through accessing the Clinical Practice Record Datalink; CPRD) and cloud-based platforms run by industry that capture detailed data on SP referrals (e.g., Access Elemental and Joy) [4–6]. These data sets offer enormous potential for understanding the impact of SP at scale, for both patients and services.

However, a major limitation to such analyses is that they rely on good quality data collection that accurately represents the experiences of patients and staff involved in SP [7]. Yet, currently, we have limited understanding of the extent to which key

data are being recorded and, as such, how 'complete' the picture presented by these datasets actually is. In other words, do analyses of administrative data represent the experiences of all patients and staff involved in SP, or do such analyses present a skewed perspective based on available records? Furthermore, if we are to improve data quality moving forwards, we also need to understand the barriers and enablers of high-quality data collection and how staff can be better supported to improve data quality.

Consequently, this analysis made use of novel data collected as part of a national survey of SP LWs to explore five interconnected questions:

1. What proportion of LWs are actively monitoring outcomes for patients and the health service and using outcome measures (e.g., ONS4) and is this compliant with the SP Information Standard?

2. What individual or contextual factors enable or hinder LWs in monitoring and recording outcomes?

3. Are the data recorded by LWs likely to be representative of the experiences of all LWs (including those who do not record)?

4. What impact monitoring approaches are taken by LWs who do monitor and what can be learnt from this for the LWs who do not monitor?

5. How can we improve monitoring of outcomes?

## Materials and methods

### Data

Data were analysed from the 2025 Social Prescribing Link Worker Survey. This survey was administered by NHS England in 2022 and 2023, and by the National Academy for Social Prescribing (NASP) in 2025. SP LWs who hold a caseload and are based in England were invited to complete the online survey. The survey focused on day-to-day role, training needs, workforce challenges, and practices of data recording and outcome measurement amongst LWs working in England.

LWs were eligible to take part if they identified as any of the following: a SP LW working in England who was recruited directly within a primary care network, recruited in secondary care, hosted within the voluntary sector and/or other community-based organisation, based within a Wider Integrated Care Board (ICB), funded by a Local Authority, or employed in another capacity but still working as a Senior SP LW Team Lead and/or Manager and holding a caseload. LWs were recruited through multiple professional and institutional channels, including NASP (via its social media, newsletter, Champions scheme and LW Advisory Group), the Future NHS platform, the Social Prescribing Network, Community of Practice contacts within Integrated Care Boards, NHSE colleagues, the Royal Society for Public Health, the Personalised Care Institute, and National Association for Voluntary and Community Action (NAVCA). It was a requirement for all participants to tick the English region and Integrated Care Board they worked in. LWs provided informed consent (aware that they could cease engagement and withdraw at any time) and those who completed the survey could select to be entered into a draw to win one of 10 £25 vouchers. The questionnaire was live from 5th February – 5th March 2025. De-identifiable data were accessed for analysis on 23rd October 2025. This study received ethical approval from University College London ethics committee (UCL-REC-2021).

### Participants

A total of 412 individuals completed the survey. As an eligibility check, the first question of the survey asked participants to describe their role. One participant was excluded from the analysis as they stated in the open text question that they were no longer a link worker. We also excluded anybody who reported being under age 18 (n = 2). This left a final analytical sample of 409. Of these, 32 chose 'prefer not to reply' to demographic questions, so these individuals were maintained in

the analyses below with the exception of regression models that relied on list-wise deletion for missing data. All other data were non-missing.

## Measures

The SP LW Survey contained 70 questions about LWs roles and experiences. This analysis focused specifically on the questions pertaining to impact monitoring and its predictors. Variables used across the analyses reported in this paper are listed below and full questions from the survey included in this paper are provided in Supplementary Methods.

**RQ1:** To assess patterns of **outcome monitoring,** LWs were asked four questions: (I) how often they record and/or monitor patient outcomes after the referral, (ii) how often they record and/or monitor impact on services after the referral, (iii) how often they record patient and/or client outcome measures (e.g., ONS4), and (iv) whether they capture outcomes through qualitative methods. Answers for (i)-(iii) were on 5-point Likert scales (never, rarely, sometimes, often, very often), while answers to (iv) were provided as tick boxes of the types of qualitative data captured (recoded as any vs none). Given there can sometimes be confusion in understanding which data are considered service-level and patient-level (e.g., whether reductions in GP appointments), for RQ2, responses to the first two questions were combined into a single variable assessing the highest frequency with which people reported monitoring impact and used this variable for further analyses.

Questions on **data-related capabilities** asked (on a 5-point Likert scale from strongly agree to strongly disagree) whether LWs had training on local clinical systems, were aware of the Social Prescribing Information Standard, were familiar with SNOMED codes, were confident in adding SNOMED codes to patient records, were able to access patient records, and were able to input into patient records.

**RQ2: Demographics** included age (in groups), gender (male, female, other/prefer not to say), ethnicity (white, other ethnic group, prefer not to say), disability (no, yes, prefer not to say), education (up to A-level, undergraduate, higher degree), previous experience working in healthcare (yes, no), and whether LWs had considered resigning in the next year (yes, no).

**Training** questions asked about their data-related capabilities (as outlined in RQ1 above) and also about whether they held a senior LW role (e.g., SP LW team lead, manager or senior SP LW), whether they receive any supervision in their role, and if they have a training budget.

**Context** questions asked about the region of England the LW worked in, whether they worked in a GP practice, whether they were funded through the Additional Roles Reimbursement Scheme (ARRS), patient caseload size (0–100, 101–200, 201–300, 301+), whether outcomes data are shared with anybody, and whether outcomes data influence finance or investment decisions.

**RQ3:** For **perceived impact**, LWs reported whether they felt their work has a **positive impact** on patients (5-point Likert scale from strongly disagree to strongly agree). More detailed questions asked on a 5-point Likert scale from strong positive impact to strong negative impact the perceived impact for various health outcomes (physical health, mental health, social connections, GP contacts, hospital contacts, number of medications used and ability to work). Comparative effectiveness of different types of interventions was reported on a 4-point Likert scale from very effective to not at all effective for age-related, arts/cultural, faith-based, and nature-based activities, as well as for healthcare and information/advice services and heritage and physical activities.

**RQ4:** LWs reported how regularly they collect outcomes from patients, who they share this information with (as in RQ2), which tools they use to measure SP outcomes, and whether they use IT systems to track and monitor referrals.

**RQ5:** LWs were asked to what extent they agree that **data-related skills** are important to their role, including understanding outcome measurement tools, confidence using data systems, and understanding the benefits of SP for health and wellbeing. Additionally, LWs were asked what they feel they need to be able to consistently capture patient outcomes more regularly.

## Analyses

Data were analysed with combinations of descriptive and inferential statistics. Any small cells (<3 participants) were combined with the nearest category to avoid potential for disclosure. For RQ2, to test if there were different predictors of monitoring outcomes, both multiple logistic regression models (comparing LWs who monitored outcomes often/very often vs other) and ordinal regression models (exploring movement across any levels of the 5-point scale) were used. To avoid potential "Table 2 fallacy", predictors were entered in blocks of related factors, each just adjusted for basic demographics [8]. Model 1 was adjusted for age, gender, and other demographic predictors (ethnicity, disability, education, work experience, resignation intention), Model 2 for age, gender and role predictors, and Model 3 adjusted for age, gender and contextual predictors. For regression models, we confirmed independence of observations through a scatterplot of the residuals. We confirmed no extreme outliers using Cook's distance for each observation. Lastly, the absence of multicollinearity was verified with all Variance Inflation Factor (VIF) values below 10. For RQ3 and RQ5, differences in responses between LWs who did and did not monitor outcomes were assessed using Pearson's Chi-square tests. Analyses were carried out in Stata v18.

## Results

### Demographic characteristics

The total sample of 409 LWs demonstrated substantial heterogeneity in terms of personal demographics and characteristics of their jobs, with every Integrated Care System in England represented (Table 1).

17% of LWs surveyed reported that their services delivered specialist SP programmes targeting one or more specific populations (Fig 1).

There was also substantial diversity in the interventions LWs reported referring patients to, with the most common being information and advice services, followed by age-related activities, healthcare services and physical activities, and the least common being faith-based and heritage activities (Fig 2).

### RQ1: Impact reporting patterns

In total, 57.5% of the sample reported recording/monitoring patient-level impact often or very often, and 45.5% reported recording/monitoring service-level impact often or very often. However, 1 in 4 LWs reported rarely or never recording patient-level impact and 1 in 3 reported rarely or never recording service-level impact (Table 2). 1 in 2 LWs reported regularly using quantitative measures, although 1 in 4 said they never use these, and 91% reported capturing qualitative data (although the frequency of this data capture and what percentage of it pertained to outcomes was not detailed).

In relation to data-related capabilities amongst LWs, only 16% strongly agreed they were aware of the SP Information Standard and only 1 in 4 was familiar with SNOMED codes (Table 3).

### RQ2: Predictors of impact reporting

When exploring what factors predicted recording/monitoring of outcomes (using the combined variable of patient and service-level impact), there was little evidence that **demographic characteristics** predicted whether LWs monitored impact or not, except for age and resignation intention (Fig 3, S1, S2 Figs, S1 –S3 Tables). LWs aged over 55 were less likely to report that they monitored outcomes (49% lower odds of reporting often/very often; OR 0.51 95%CI 0.27–0.97). And people who were considering resigning in the next year were 54% less likely to report monitoring outcomes (OR 0.46 95%CI 0.29–0.71).

There was also little evidence that **role-related factors** predicted monitoring of outcomes. Neither a LW's seniority, whether they had supervision, or whether they had a training budget predicted outcome monitoring. However, LWs who were aware of the SP Information Standard were 2.3 times more likely to report outcomes (95%CI 1.53–3.42).

**Table 1. Sample characteristics.**

| | N (%) |
|---|---|
| Personal characteristics | |
| Age group | |
| 18-24 | 9 (2.2%) |
| 25-34 | 62 (15.2%) |
| 35-44 | 86 (21.0%) |
| 45-54 | 124 (30.3%) |
| 55-64 | 102 (24.9%) |
| 65+ | 14 (3.4%) |
| Prefer not to say | 12 (2.9%) |
| Gender | |
| Male | 50 (12.2%) |
| Female | 350 (85.6%) |
| Other/ prefer not to say | 9 (2.2%) |
| Ethnicity | |
| White | 326 (79.7%) |
| Other ethnic group | 58 (14.2%) |
| Prefer not to say | 25 (6.1%) |
| Disability | |
| No | 363 (88.8%) |
| Yes | 43 (10.5%) |
| Prefer not to say | 3 (0.7%) |
| Education | |
| 1. Up to A-level/BTEC nationals/advanced apprenticeship | 145 (35.5%) |
| 2. Undergraduate degree/foundation degree/ higher apprenticeship | 194 (47.4%) |
| 3. Master's degree/PhD | 70 (17.1%) |
| Worked previously in healthcare | |
| No | 292 (71.4%) |
| Yes | 117 (28.6%) |
| JOB CHARACTERISTICS | |
| Job role | |
| 1. SPLW (Primary Care Network) | 213 (52.1%) |
| 2. SPLW (Voluntary, community, faith, and social enterprise/VCFSE) | 104 (25.4%) |
| 3. SPLW (General Practitioner/GP Federation) | 44 (10.8%) |
| 4. SPLW (Local Authority) | 29 (7.1%) |
| 5. SPLW (NHS Trust) | 9 (2.2%) |
| 6. SPLW (Housing Organisation) | 3 (0.7%) |
| 7. Other | 7 (1.7%) |
| Region | |
| 1. East of England | 41 (10.0%) |
| 2. London | 58 (14.2%) |
| 3. Midlands | 71 (17.4%) |
| 4. North East & Yorkshire | 58 (14.2%) |
| 5. North West | 63 (15.4%) |

*(Continued)*

| | N (%) |
|---|---|
| 6. South East | 67 (16.4%) |
| 7. South West | 51 (12.5%) |
| Patient caseload | |
| 1. 0-100 | 72 (17.6%) |
| 2. 101-200 | 109 (26.7%) |
| 3. 201-300 | 143 (35.0%) |
| 4. 301+ | 85 (20.8%) |
| Primary referral source | |
| 1. GPs | 254 (62.1%) |
| 2. Primary Care Network/PCN staff | 114 (27.9%) |
| 3. self-referrals | 12 (2.9%) |
| 4. Social services | 6 (1.5%) |
| 5. Secondary care | 3 (0.7%) |
| 6. Other (schools, Child and Adolescent Mental Health Service/CAMHS, VCFSE etc) | 20 (4.9%) |
| Funded through the Additional Roles Reimbursement Scheme (ARRS) | |
| No | 176 (43.0%) |
| Yes | 233 (57.0%) |
| Works from GP practice | |
| No | 157 (38.4%) |
| Yes | 252 (61.6%) |
| Receives some supervision | |
| No | 26 (6.4%) |
| Yes | 383 (93.6%) |

There was more evidence that **contextual factors** had an impact on outcome monitoring. If the outcomes were shared with somebody (e.g., clinicians, funders, patients or other stakeholders), LWs were 6 times more likely to monitor them regularly (95%CI 2.93–12.43). If the outcomes data were reportedly used to inform investment decisions, LWs were also 1.8 times more likely to monitor outcomes regularly (95%CI 1.03–3.02).

As the main variable on recording/monitoring impact did not specify whether this monitoring was written down anywhere (and thus available for others to see), we also repeated the analysis looked specifically at whether LWs recorded outcome measures (e.g., ONS4). Notably, knowledge of the SP Information Standard was not a predictor of using outcomes measures, although sharing the outcomes with somebody and the outcomes being used to inform investment decisions were even larger predictors (OR 7.2, 95%CI 2.36–21.76 and OR 2.59, 95%CI 1.29–5.22 respectively for binary outcomes). But we also found that LWs were 78% *less* likely to use outcome measures if they accessed patient medical records (95% 0.11–0.41) (Fig 4, S3 Fig, S4 and S5 Tables).

### RQ3: Potential differences between LWs who captured and not captured outcomes data

There were high levels of reported belief that SP had a positive impact on patients both amongst those who did and did not report using outcome measures or monitoring outcomes (Fig 5; exploratory descriptive breakdowns of reporting depending on type of outcome and type of impact reporting are shown in S6-S9 Tables). Once adjusting for demographic, training and context confounders, there was no evidence of a difference in beliefs of positive effects amongst those who

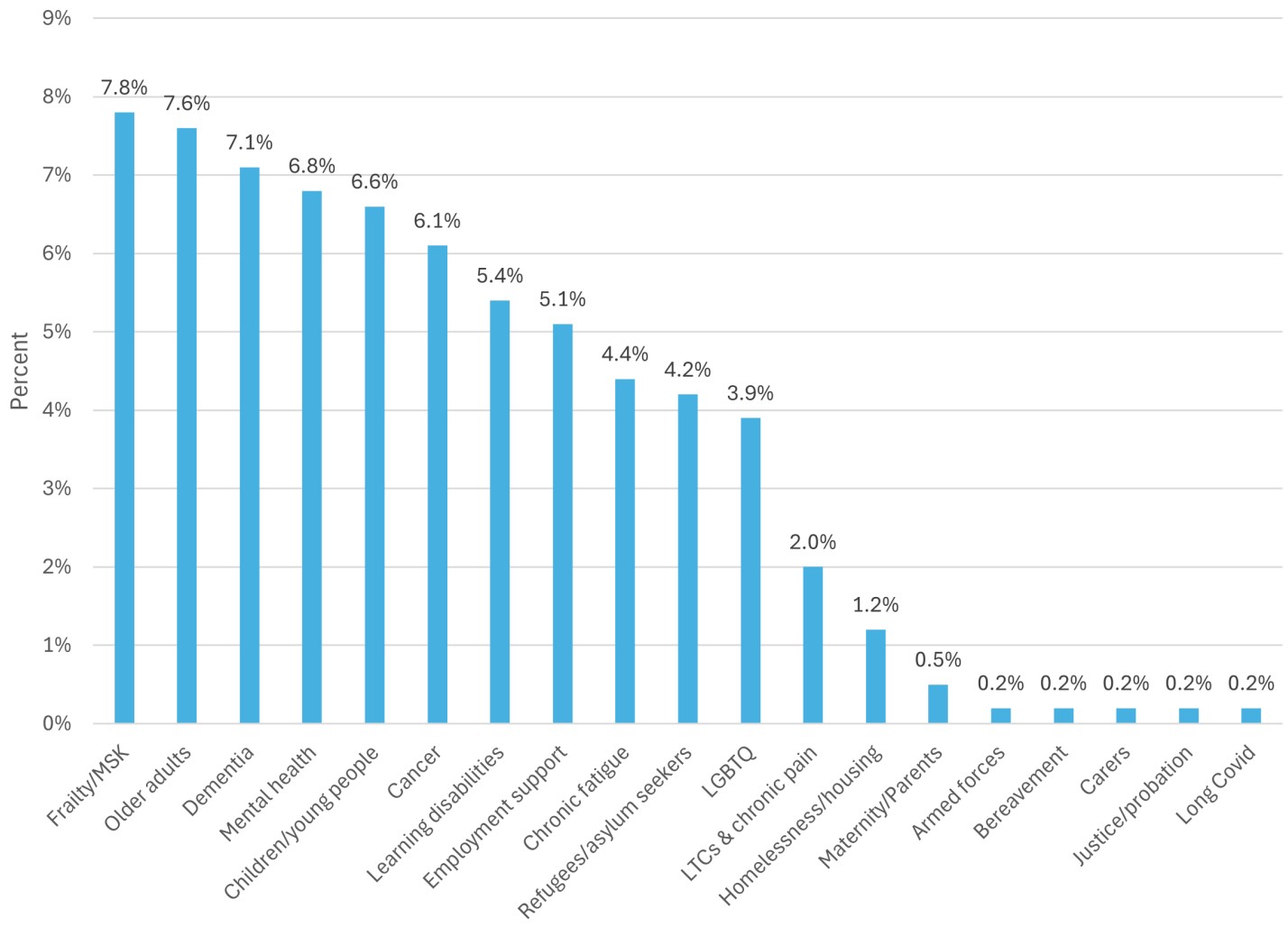

**Fig 1. Specialist social prescribing services.** *Note: more than one option could be selected, providing overall percentages that can exceed 100.*

did and did not monitor outcomes (OR 0.71, 95% CI 0.43–1.19). Indeed, when using the variable of whether people report outcomes using scales such as ONS4, those who recorded outcomes in this way were 45% less likely to believe that SP had a positive impact on patients (OR 0.55, 95% CI 0.32–0.94).

Of those who did not use quantitative patient outcome measures (e.g., ONS4), 80.4% of them did capture qualitative data using combinations of case studies, patient stories, patient feedback forms, and informal feedback from patients, carers, and VCFSOs (Fig 6).

## RQ4: Learnings from LWs who do use patient outcome measures

Of the LWs who did record patient outcome measures, 60% collected data at the beginning and end of interactions with patients, with flexibility depending on how long their interaction with their patients was (Table 4). 1 in 5 gathered data at every single interaction. 85% of those who did collect outcomes reported sharing them with somebody (e.g., clinicians, funders, or patients).

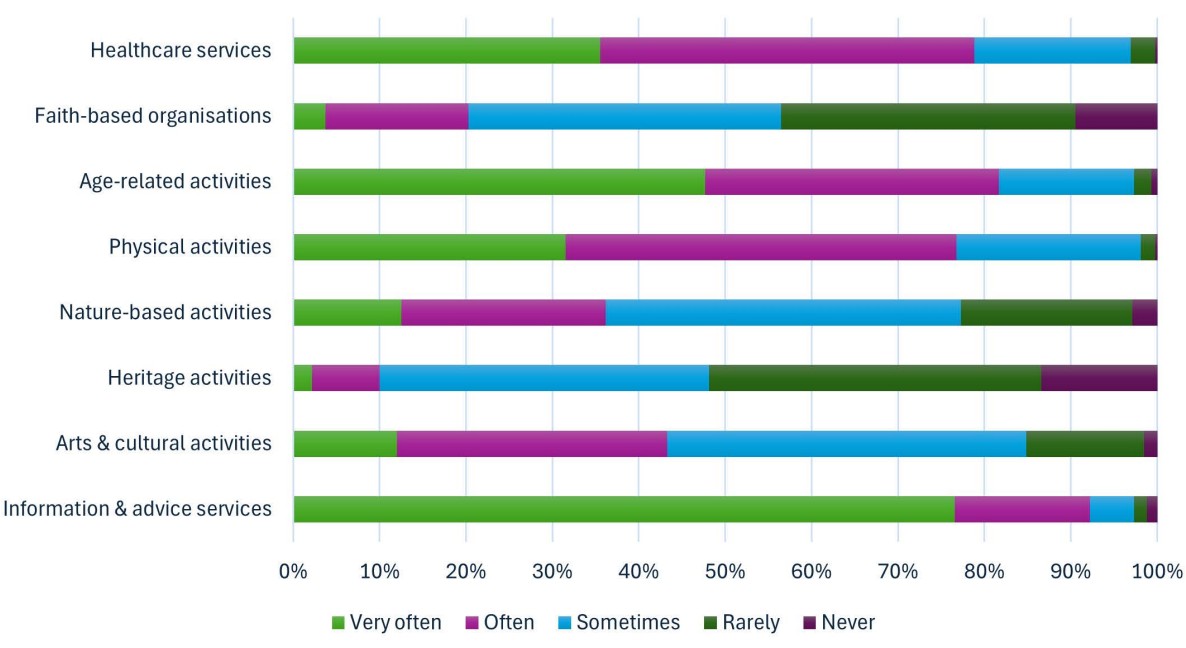

**Fig 2. Frequency of referrals to different interventions.**

**Table 2. Patterns of impact reporting (N = 409).**

|  | Frequency (%) |
| --- | --- |
| Records/monitors patient impact |  |
| 1. Very often | 128 (31.3%) |
| 2. Often | 107 (26.2%) |
| 3. Sometimes | 76 (18.6%) |
| 4. Rarely | 44 (10.8%) |
| 5. Never | 54 (13.2%) |
| Records/monitors service impact |  |
| 1. Very often | 105 (25.7%) |
| 2. Often | 81 (19.8%) |
| 3. Sometimes | 87 (21.3%) |
| 4. Rarely | 57 (13.9%) |
| 5. Never | 79 (19.3%) |
| Records outcome measures (e.g., ONS4) |  |
| 1. Very often | 94 (23.0%) |
| 2. Often | 105 (25.7%) |
| 3. Sometimes | 73 (17.8%) |
| 4. Rarely | 39 (9.5%) |
| 5. Never | 98 (24.0%) |
| Captures qualitative data |  |
| Yes | 372 (91.0%) |
| No | 37 (9.0%) |

 

**Table 3. Data-related capabilities (N = 409).**

| | Frequency (%) |
|---|---|
| Had training on local clinical system | |
| 1. Strongly agree | 120 (29.3%) |
| 2. Agree | 162 (39.6%) |
| 3. Neither agree nor disagree | 33 (8.1%) |
| 4. Disagree | 39 (9.5%) |
| 5. Strongly disagree | 55 (13.4%) |
| Aware of Social Prescribing Information Standard | |
| 1. Strongly agree | 67 (16.4%) |
| 2. Agree | 141 (34.5%) |
| 3. Neither agree nor disagree | 79 (19.3%) |
| 4. Disagree | 85 (20.8%) |
| 5. Strongly disagree | 37 (9.0%) |
| Familiar with SNOMED codes | |
| 1. Strongly agree | 99 (24.2%) |
| 2. Agree | 125 (30.6%) |
| 3. Neither agree nor disagree | 59 (14.4%) |
| 4. Disagree | 73 (17.8%) |
| 5. Strongly disagree | 53 (13.0%) |
| Confident adding SNOMED codes to patient records | |
| 1. Strongly agree | 97 (23.7%) |
| 2. Agree | 78 (19.1%) |
| 3. Neither agree nor disagree | 77 (18.8%) |
| 4. Disagree | 85 (20.8%) |
| 5. Strongly disagree | 72 (17.6%) |
| Able to access patient records | |
| 1. Strongly agree | 238 (58.2%) |
| 2. Agree | 87 (21.3%) |
| 3. Neither agree nor disagree | 7 (1.7%) |
| 4. Disagree | 26 (6.4%) |
| 5. Strongly disagree | 51 (12.5%) |
| Able to input into patient records | |
| 1. Strongly agree | 256 (62.6%) |
| 2. Agree | 92 (22.5%) |
| 3. Neither agree nor disagree | 10 (2.4%) |
| 4. Disagree | 20 (4.9%) |
| 5. Strongly disagree | 31 (7.6%) |

Amongst those who specifically used patient outcome measures, the most common measurement tool used was ONS4 (used by 61% of LWs who did monitor outcomes; Fig 6 top panel), with most LWs recording their outcomes directly in electronic health records (e.g., EMIS, SystmOne, Rio etc). Of the digital systems specifically designed to record SP, the platform used most often was Joy (21%) followed by Elemental (7%) and Social Rx (5%) (Fig 7 bottom panel).

## RQ5: Solutions to improve the monitoring of outcomes

Between those who did and did not report monitoring impact, there were differences in the perceived importance of having skills and understanding in certain domains. Those who monitored impact regularly reported feeling that knowledge

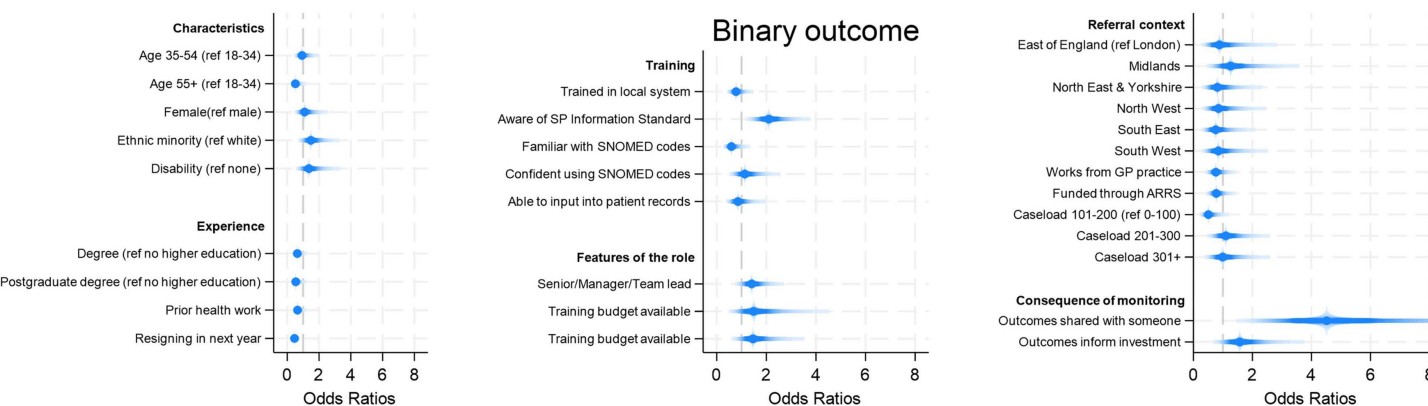

**Fig 3. Predictors of recording/monitoring impact (using any means).** *Notes: Binary outcome considers which factors predict people reporting that they monitor often/very often vs other. Age and gender were included in all models.*

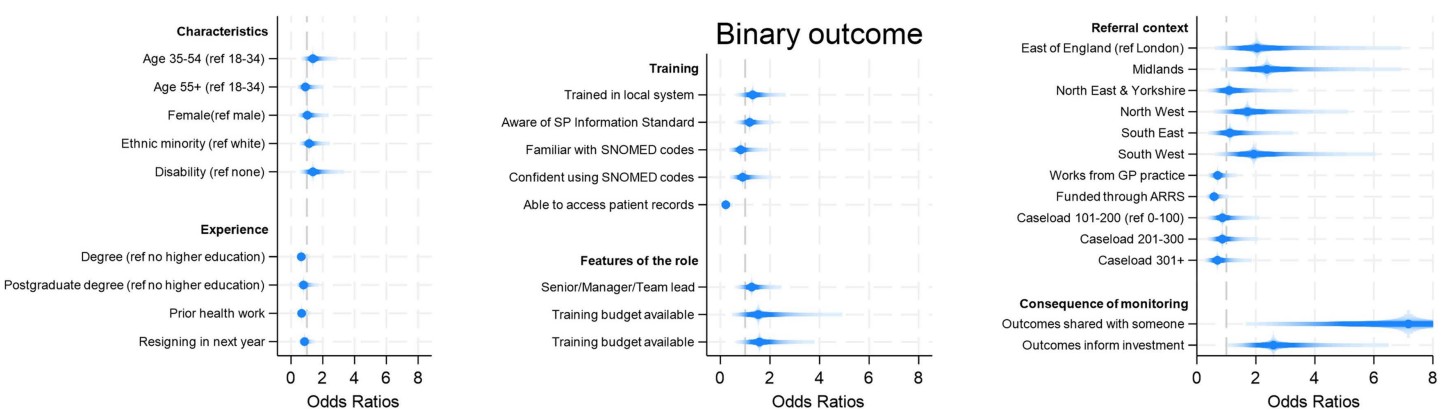

**Fig 4. Predictors of using patient outcomes measures (e.g., ONS4).** *Notes: Binary outcome considers which factors predict people reporting that they monitor often/very often vs other. Age and gender were included in all models.*

of outcome measurement tools was more important than those who do not monitor (85.9% agree or strongly agree vs 65.1%; Table 5). LWs who specifically used patient outcome measures reported very similar views (86.0% agree/strongly agree vs 67.1%; Table 6). However, there were some discrepancies for other perceived data-related skills. For monitoring impact more broadly, LWs who did monitor felt that being confident in using data systems (including data capture and data reporting) was important (91.4% agree/strongly agree vs 76.2%), and they also reported stronger agreement that knowledge of the health benefits of SP was important (69.5% strongly agree vs 51.3%). But there was no such difference in feelings amongst those who specifically use patient outcome measures compared to those who do not.

LWs who did not monitor outcomes were asked what would improve their impact monitoring (Fig 8). Simpler processes and better digital tools were highest in the requests, alongside more advanced clinical record integration. 1 in 3 LWs reported they would value training on how to use outcome measures. Other LWs suggested more time during appointments to capture outcomes (17%). These requests were very similar to those from LWs who already used outcome measures, with this group showing even higher calls for this support (28%) and asking for more advanced clinical record integration (40%). Notably, however, nearly 14% LWs who did not report outcomes did not feel that patient outcome measures should be used, with 18% of those who did use them sharing this sentiment.

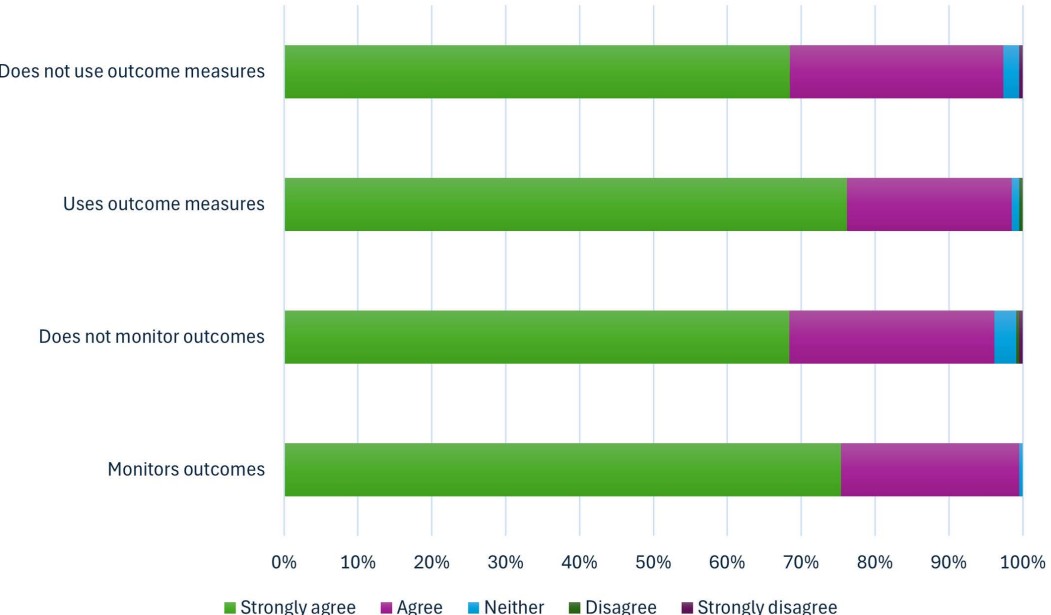

**Fig 5. Belief that SP has a positive impact on patients depending on whether LWs monitor impact or use quantitative outcome measures.**

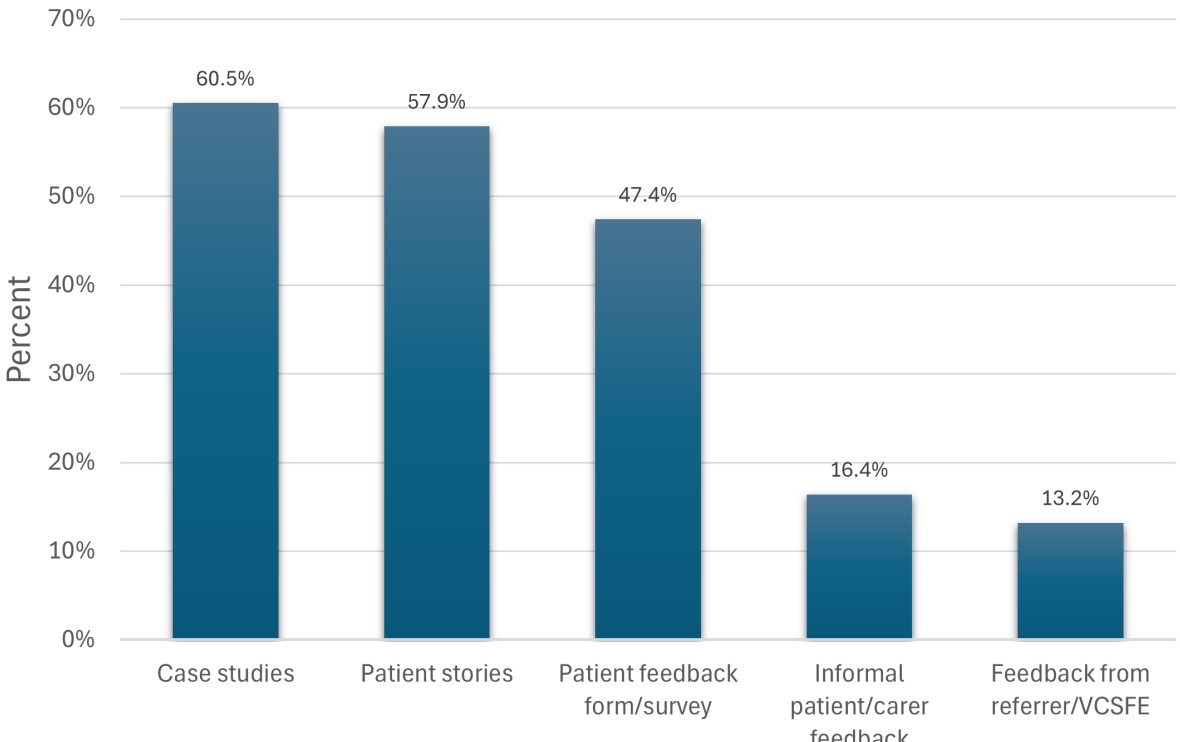

**Fig 6. Types of qualitative data captured.** *Note: more than one option could be selected, providing overall percentages that can exceed 100.*

**Table 4. Impact reporting behaviours amongst those who do use patient outcome measures.**

|  | Record outcome measures N (%) |
|---|---|
| Timing of outcome recording | |
| 1. At every interaction | 21 (20.4%) |
| 2. Beginning and end | 62 (60.2%) |
| 3. Every 6 weeks | 13 (12.6%) |
| 4. Every 12 weeks or less | 7 (7.8%) |
| Outcomes shared with somebody | |
| No | 32 (15.2%) |
| Yes | 178 (84.8%) |
| N = 220 | |

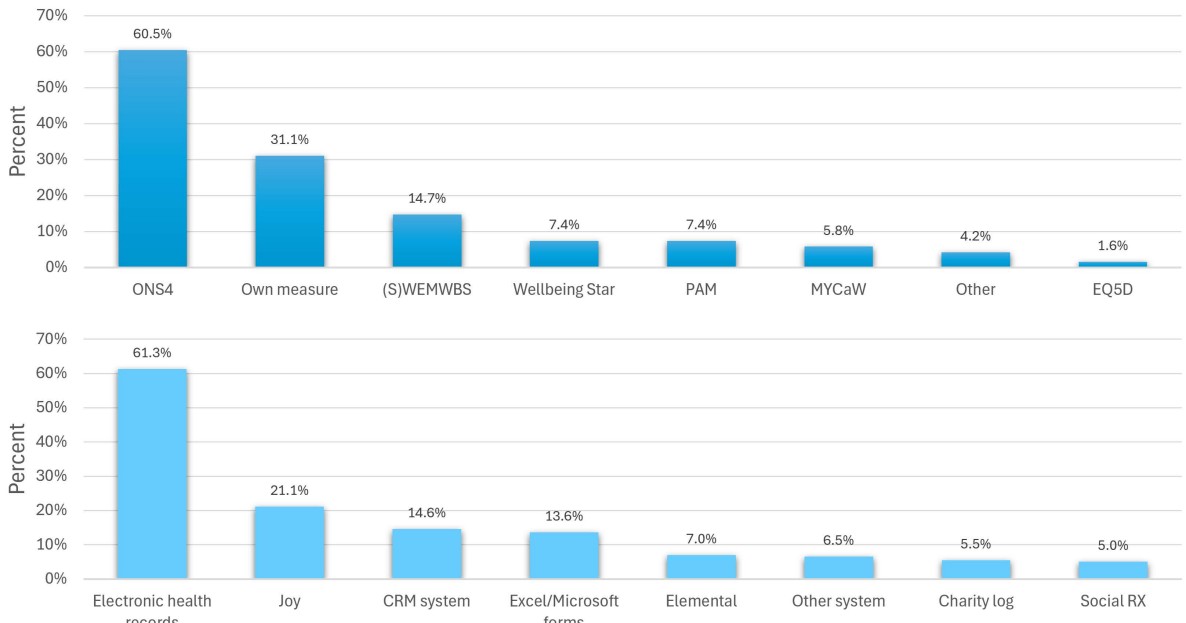

**Fig 7. Measurement tools and systems used for data capture.** *Note: more than one option could be selected, providing overall percentages that can exceed 100.*

## Discussion

This set of analyses aimed to enrich our understanding of current rates, patterns and predictors of outcomes monitoring by SP LWs in England, what the consequences might be to our understanding of the impact of SP, and how outcomes monitoring could be improved moving forwards. We looked at two main outcomes that are closely related: (i) monitoring the impact of SP following referral to an intervention (using any approach, qualitative or quantitative and irrespective of how this monitoring was done), and (ii) specifically using quantitative patient outcome measures (e.g., ONS4). Most importantly, there was clear evidence that outcomes data are not being monitored and reported according to the SP Information Standard minimum dataset requirements. Only around 1 in 2 LWs is monitoring patient or service-level outcomes regularly, and fewer than half are regularly using quantitative outcome measures such as ONS4. This is an improvement from 2023: 23% now say they very often record quantitative outcomes vs just 16% reported in 2023, and

**Table 5. Perceived importance of data-related skills amongst those who monitor impact.**

| | Not monitoring | Monitoring | Total | Chi-square test |
|---|---|---|---|---|
| Outcome measurement tools | | | | |
| 1. Strongly agree | 31 (16.4%) | 66 (30.0%) | 97 (23.7%) | <0.001 |
| 2. Agree | 92 (48.7%) | 123 (55.9%) | 215 (52.6%) | |
| 3. Neither agree nor disagree | 50 (26.5%) | 22 (10.0%) | 72 (17.6%) | |
| 4. Disagree/ strongly disagree | 16 (8.4%) | 9 (4.1%) | 25 (6.1%) | |
| Data capture, reporting & using data systems | | | | |
| 1. Strongly agree | 47 (24.9%) | 86 (39.1%) | 133 (32.5%) | <0.001 |
| 2. Agree | 97 (51.3%) | 115 (52.3%) | 212 (51.8%) | |
| 3. Neither agree nor disagree | 37 (19.6%) | 14 (6.4%) | 51 (12.5%) | |
| 4. Disagree/ strongly disagree | 8 (4.2%) | 5 (2.3%) | 13 (3.2%) | |
| Health benefits of SP | | | | |
| 1. Strongly agree | 97 (51.3%) | 153 (69.5%) | 250 (61.1%) | 0.002 |
| 2. Agree | 89 (47.1%) | 65 (29.5%) | 154 (37.7%) | |
| 3. Neither agree nor disagree | 2 (1.1%) | 0 (0.0%) | 2 (0.5%) | |
| 4. Disagree/ strongly disagree | 1 (0.5%) | 2 (1.0%) | 3 (0.7%) | |
| N = 409 | | | | |

**Table 6. Perceived importance of data-related skills who use patient outcome measures specifically.**

| | Not monitoring | Monitoring | Total | Chi-square test |
|---|---|---|---|---|
| Outcome measurement tools | | | | |
| 1. Strongly agree | 37 (17.6%) | 60 (30.2%) | 97 (23.7%) | <0.001 |
| 2. Agree | 104 (49.5%) | 111 (55.8%) | 215 (52.6%) | |
| 3. Neither agree nor disagree | 51 (24.3%) | 21 (10.6%) | 72 (17.6%) | |
| 4. Disagree/ strongly disagree | 18 (8.6%) | 7 (3.5%) | 25 (6.1%) | |
| Data capture, reporting & using data systems | | | | |
| 1. Strongly agree | 65 (31.0%) | 68 (34.2%) | 133 (32.5%) | 0.899 |
| 2. Agree | 110 (52.4%) | 102 (51.3%) | 212 (51.8%) | |
| 3. Neither agree nor disagree | 27 (12.9%) | 24 (12.1%) | 51 (12.5%) | |
| 4. Disagree/ strongly disagree | 8 (3.8%) | 4 (2.5%) | 13 (3.2%) | |
| Health benefits of SP | | | | |
| 1. Strongly agree | 119 (56.7%) | 131 (65.8%) | 250 (61.1%) | 0.104 |
| 2. Agree | 90 (42.9%) | 64 (32.2%) | 154 (37.7%) | |
| 3. Neither agree nor disagree | 1 (0.5%) | 1 (0.5%) | 2 (0.5%) | |
| 4. Disagree/ strongly disagree | 0 (0.0%) | 3 (1.5%) | 3 (0.7%) | |
| N = 409 | | | | |

only 24% say they never do this compared to 28% in 2023 [9]. But the progress is modest for a two-year period. Contrary to the technical recommendations from NHS England that outcomes data collection should be every six months, there is substantial variability in the timing of outcome recording [3]. Further, although LWs are asked in NHS guidance to use the ONS4 Wellbeing Scale, of those who do monitor outcomes, only 61% report using this scale [3]. However, it is encouraging that the other scales frequently used are those recommended by the NHS such as MyCaW, PAM and Wellbeing Star [10]. It is also noteworthy that the scales themselves were not perceived as a barrier to impact monitoring, with only 6% of those who do not use patient outcome measures regularly saying that they wish for better measurement scales. A

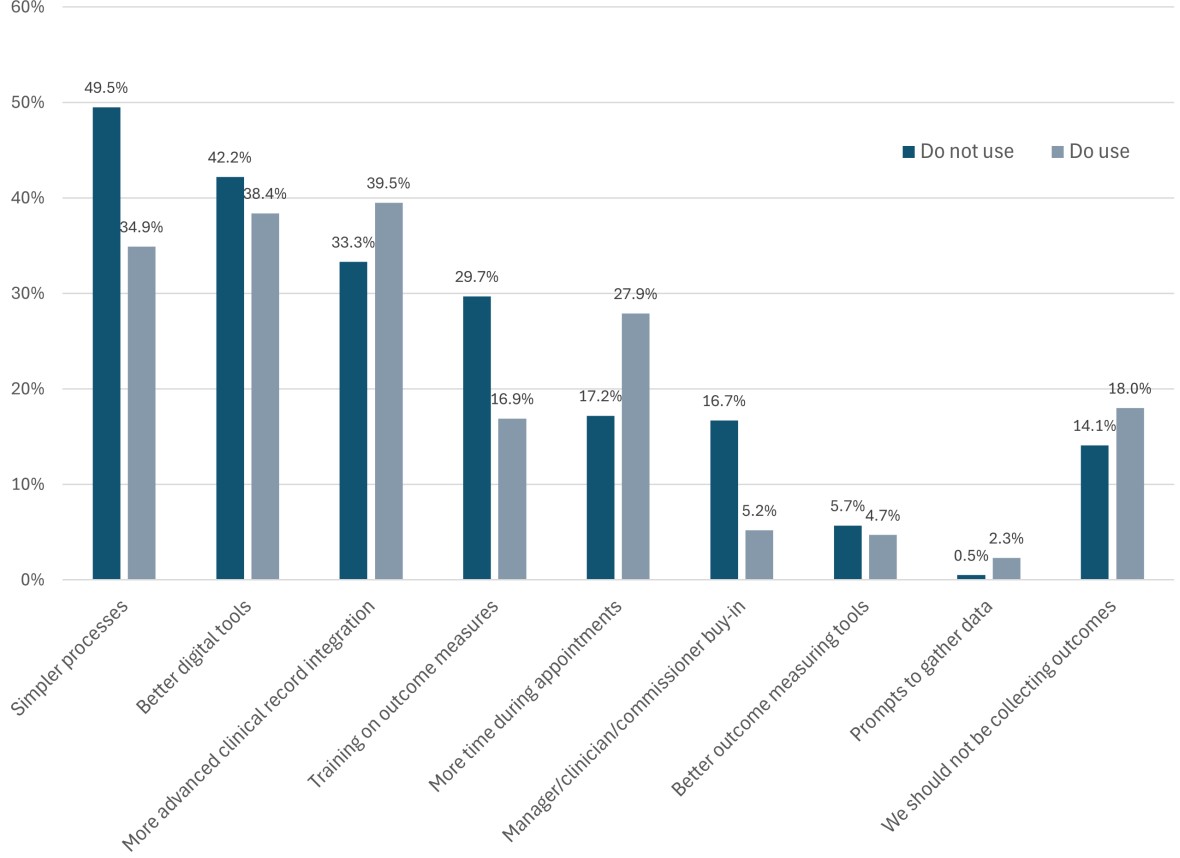

**Fig 8. Proposed solutions to improve use of patient outcome measure (e.g., ONS4).** *Note: more than one option could be selected, providing overall percentages that can exceed 100.*

further concern, though, is that 14% of LWs reported capturing outcomes data in closed systems such as Excel/Microsoft Forms that cannot be analysed beyond the practice/setting where they are recorded unless they are proactively shared externally, disabling routine nation-wide audits. Overall, we found no evidence that the outcomes data on overall impact of SP 'missing' in large-scale analyses (such as those that use CPRD or Elemental) is less positive from the data currently being reported. Indeed, those LWs who did not report recording outcomes using scales such as ONS4 were slightly more likely to report believing that SP had positive outcomes, suggesting there may be an under-reporting of patient benefit. However, perceived benefits do not necessarily equate to changes on validated measures, so a more conservative interpretation is that we have no further reason to suggest that 'missing' data present a more negative picture of patient impact. There were some slight differences between LWs who did and did not monitor outcomes in perceptions about how patients were affected by SP and which types of interventions were most effective. But these analyses were exploratory and descriptive. It remains possible that other factors that could have affected these perceptions such as dissatisfaction with the job role. Nonetheless, it is important moving forwards that fuller data on SP outcomes is captured and reported.

In considering why there are still relatively low levels of routine impact monitoring, we found no meaningful evidence that this is due to the backgrounds or training of LWs. However, we did see that LWs who are considering resigning are less likely to report impact, which could suggest a loss of motivation or interest in the role, especially for an activity like impact monitoring that may be perceived as secondary to the core work of patient interactions. The clearest predictors of

impact monitoring were awareness of the SP Information Standard (2.3 times more likely to monitor), sharing the outcomes with somebody (6 times more likely) and having them used to inform investment decisions (1.8 times more likely). When we looked specifically at predictors of using patient outcome measures like ONS4, we actually found that people with access to clinical records were *less* likely to use them, which could be because they assume outcomes are being automatically recorded through the capture of other routine information about patients' outcomes, or potentially because of challenges with the integration of outcome monitoring tools with electronic patient records (something raised by 40% of LWs who do use outcome measures). But sharing the outcomes with somebody and them being used to inform investment decisions were still important predictors. It is likely that all of these predictors increase motivation (or even pressure) to record outcomes as it is perceived to be a core part of the role necessary for the service to continue. For example, it might not so much by having somebody with whom to share the outcomes data that increases outcomes reporting, but rather having somebody directly *asking* for the outcomes data that creates this pressure and accountability to monitor outcomes. These results resonate with the finding that those who do monitor outcomes value the skills relating to data capture, reporting and understanding of the impacts more highly, suggesting an overall acceptance of the centrality of outcomes monitoring to the role of a LW.

In considering how the perceived importance of outcomes monitoring could be raised, one potential modifiable factor relates to the SP Information Standard. Knowledge of this was related to increased impact monitoring, possibly because this created a similar sense of expectation around monitoring as having a member of staff in the organisation asking to see the outcomes data. But only 51% reported being agreeing or strongly agreeing that they were aware of the SP Information Standard and 55% with SNOMED codes, which compares with 55% and 59% respectively in 2023. The question was asked as a binary in 2023's survey, precluding direct comparisons between these figures, but it still suggests little improvement in the two years. While these analyses cannot prove causality, they do suggest a potential value in trying to raise awareness of the SP Information Standard amongst LWs to see if this improves outcome reporting. One point of note is that the Standard is complex to read and involves multiple parts. Given that nearly half of LWs (including those who do already monitor outcomes) requested simpler processes, creating a simplified guide could be a viable intervention. And its potential to increase outcomes reporting can be assessed in future SPLW surveys. The good practice recorded from the LWs who do monitor outcomes suggest that ONS4 is the most commonly used scale and reporting of outcomes in electronic health records is most common. Both of these approaches are strong in providing data that can be accessed and analysed by researchers through platforms such as EMIS. Given that 2 in 5 LWs said they wished they had better digital tools, providing simplified guidance on using electronic health records could also be valuable to show how they can be used for monitoring outcomes. For LWs outside of clinical contexts, systems such as Joy and Elemental also received some indications of regular use (unlike other systems like Social RX, which was not frequently reported). Given Joy and Elemental can both be analysed to understand aggregate outcomes as well as capturing additional more detailed information on referrals, there could be a value to recommending the use of them or other similar platforms in simplified SP information standard guidance.

These analyses are limited by only reporting on a relatively small sample of LWs, which we estimate to be around 11% of all LWs in the country. While assessing the representativeness of this sample of LWs is not possible given no official national data on LWs, the geographical and demographical diversity of responses suggests that the survey did capture heterogeneous experiences that make the data meaningful when considering the experiences and behaviours of LWs as a collective. However, it is important to interpret the findings in the context of the fact that they do not represent the views of the entire population link workers. Future surveys should aim to recruit more LWs and attempt to assess how the sample compares to core demographics across the sector as a whole to improve the inferences that can be made from the data. The analyses presented here were also limited by the questions available. Future surveys are encouraged to ask specific follow-up questions on data and related activities that would improve data interpretation, such as how often LWs meet with patients, what proportion are referred to an activity, and how they record these varying implementation factors within their outcomes reporting. Nonetheless, the results provided here provide some valuable benchmarking against

which future interventions aimed at improving outcomes monitoring and the use of patient outcome measures specifically can be evaluated.

## Supporting information

**S1 File. Supplementary methods.**
(DOCX)

**S1 Fig. Correlation matrix of variables included in regression analyses.**
(PNG)

**S2 Fig. Predictors of recording/monitoring impact (using any means).** Notes: Ordinal outcomes consider which factors predict movement across any levels of the 5-point scale. Age and gender were included in all models.
(JPG)

**S3 Fig. Predictors of using patient outcomes measures (e.g., ONS4).** Notes: Ordinal outcomes consider which factors predict movement across any levels of the 5-point scale. Age and gender were included in all models.
(JPG)

**S1 Table. Sample characteristics by impact monitoring.**
(DOCX)

**S2 Table. Logistic regression model for monitoring outcomes often or very often; odds ratios and confidence intervals.**
(DOCX)

**S3 Table. Ordinal regression model for monitoring outcomes; odds ratios and confidence intervals.**
(DOCX)

**S4 Table. Logistic regression model for recording quantitative outcomes (e.g., ONS4) often or very often; odds ratios and confidence intervals.**
(DOCX)

**S5 Table. Ordinal regression model for recording quantitative outcomes (e.g., ONS4); odds ratios and confidence intervals.**
(DOCX)

**S6 Table. Perceived impacts by impact monitoring/reporting.**
(DOCX)

**S7 Table. Perceived impacts by use of patient outcome measures.**
(DOCX)

**S8 Table. Perceived effectiveness of different interventions by impact.**
(DOCX)

**S9 Table. Perceived effectiveness of different interventions by use of patient outcome measures.**
(DOCX)

## Author contributions

**Conceptualization:** Daisy Fancourt.

 

**Formal analysis:** Daisy Fancourt.

**Funding acquisition:** Daisy Fancourt, Feifei Bu.

**Investigation:** Daisy Fancourt.

**Methodology:** Daisy Fancourt.

**Software:** Daisy Fancourt.

**Validation:** Feifei Bu.

**Visualization:** Daisy Fancourt.

**Writing – original draft:** Daisy Fancourt.

**Writing – review & editing:** Daisy Fancourt, Feifei Bu.

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
