## [Decision Letter · Decision Letter 0]

18 Feb 2026

Dear Dr. Fancourt,

Thank you for submitting your manuscript to PLOS ONE. After careful consideration, we feel that it has merit but does not fully meet PLOS ONE’s publication criteria as it currently stands. Therefore, we invite you to submit a revised version of the manuscript that addresses the points raised during the review process.

**ACADEMIC EDITOR:** **Please review the reviewer comments. Your manuscript comes well recommended and needs some minor revisions prior to acceptance. Please revise the manuscript and resubmit for further consideration.**

We look forward to receiving your revised manuscript.

Kind regards,

Souparno Mitra, M.D.

Academic Editor

PLOS One

Reviewers' comments:

Reviewer's Responses to Questions

**Comments to the Author**

1. Is the manuscript technically sound, and do the data support the conclusions?

Reviewer #1: Yes

Reviewer #2: Yes

2. Has the statistical analysis been performed appropriately and rigorously?

Reviewer #1: Yes

Reviewer #2: I Don't Know

3. Have the authors made all data underlying the findings in their manuscript fully available?

Reviewer #1: Yes

Reviewer #2: Yes

4. Is the manuscript presented in an intelligible fashion and written in standard English?

Reviewer #1: Yes

Reviewer #2: Yes

Reviewer #1: This is a secondary analysis of a cross-sectional study by the National Academy for Social Prescribing that was conducted among Social Prescribing Link Workers in Great Britain. The research questions are of high relevance to a wider readership. In principle, the methodology is adequate and the conclusions are justified. However, the sampling might result in important flaws. If more information would be available this could improve transparency and trustworthiness in the results.

Further comments:

- l. 16 methods could be a bit more informative

- l. 32f.: the Conclusions are not really related to the results section. It becomes clearer when reading the discussion, but still it could be a good idea to frame it more in the context of the results.

- l. 55 could profit from citations that offer insights into shortcomings of data quality for SP evaluations (i.e. 10.5334/ijic.6472)

- l. 90 should not include results (i.e. n) but rather explain the method of participant acquisition. Therefore, the chapter title could be moved to line 77 and l. 90 f. should be moved to the results section. The chapter should be augmented with information on the overall sample size to assess what the response rate was in the results section.

- Table 1: A legend with explanations for abbreviations should be included

- Tables 2 and 3 could be more concise, as Likert scale is the same.

- Rewrite to OR X.X (95% CI) instead of %, i.e. as in the abstract

- l. 184 many should be removed.

- l. 286 f.: I agree, still l. 210 f. is a highly interesting finding and should be further explored.

- Ordinal regression could be moved to the supplement. I do not see the benefit of having it in the main manuscript.

- Colour usage in the figures could be optimized, especially for people with visual impairments (i.e. Figure 5).

In a future iteration, the amount of time spent on documentation could be an interesting aspect.

Reviewer #2: I have reviewed and critiqued some limitations of the study "Patterns and predictors of outcome monitoring amongst link workers: learnings from the national Social Prescribing Link Worker Survey 2025." Please make necessary edits.

.

Reviewer #1: **Yes:** Hendrik NapieralaHendrik NapieralaHendrik NapieralaHendrik Napierala

Reviewer #2: **Yes:** Arun PrasadArun PrasadArun PrasadArun Prasad

---

## [Author Response · Author response to Decision Letter 1]

25 Feb 2026

Dear Editor,

We’re pleased the paper has received positive reviews. We have responded to all the changes requested below.

Best wishes,

The Authors

***The paper has been reformatted accordingly.

***We have confirmed that consent was given online. We already clarify that the data were de-identified.

***We have augmented the data availability statement to clarify these details.

***We are unaware of any retracted articles in our reference list.

Reviewer's Responses to Questions

Comments to the Author

Reviewer #1: This is a secondary analysis of a cross-sectional study by the National Academy for Social Prescribing that was conducted among Social Prescribing Link Workers in Great Britain. The research questions are of high relevance to a wider readership. In principle, the methodology is adequate and the conclusions are justified. However, the sampling might result in important flaws. If more information would be available this could improve transparency and trustworthiness in the results.

***We are pleased the reviewer feels the paper is so relevant. We have described the sampling in detail in the methods section but have also now provided an additional sentence on this in the limitations paragraph.

Further comments:

- l. 16 methods could be a bit more informative

***We have now provided more methods detail in the abstract.

- l. 32f.: the Conclusions are not really related to the results section. It becomes clearer when reading the discussion, but still it could be a good idea to frame it more in the context of the results.

***Thank you for this point. We have provided more linkages between the abstract conclusions and results.

- l. 55 could profit from citations that offer insights into shortcomings of data quality for SP evaluations (i.e. 10.5334/ijic.6472)

***We have added the suggested reference.

- l. 90 should not include results (i.e. n) but rather explain the method of participant acquisition. Therefore, the chapter title could be moved to line 77 and l. 90 f. should be moved to the results section. The chapter should be augmented with information on the overall sample size to assess what the response rate was in the results section.

***This information is not results but rather a description of how we selected the analytical sample. The overall sample size was 412, so there are no other details on response rate that can be provided.

- Table 1: A legend with explanations for abbreviations should be included

***We have now provided all acronyms in full form within the table

- Tables 2 and 3 could be more concise, as Likert scale is the same.

***We considered horizontal tables, but found vertical ones more intuitive for these results

- Rewrite to OR X.X (95% CI) instead of %, i.e. as in the abstract

***We have added the ORs now.

- l. 184 many should be removed.

***Removed

- l. 286 f.: I agree, still l. 210 f. is a highly interesting finding and should be further explored.

***We agree this is an interesting point, but we are wary about over-interpreting at this stage. We’ve already provided half a paragraph on this point so think it is prudent not to interpret further at this stage but rather to test this finding further in future studies where we can probe with additional questions.

- Ordinal regression could be moved to the supplement. I do not see the benefit of having it in the main manuscript.

***We have now made this change.

- Colour usage in the figures could be optimized, especially for people with visual impairments (i.e. Figure 5).

***We have altered the colours to make them starker.

In a future iteration, the amount of time spent on documentation could be an interesting aspect.

***Thank you for this suggestion for future iterations of the survey!

Reviewer #2: I have reviewed and critiqued some limitations of the study "Patterns and predictors of outcome monitoring amongst link workers: learnings from the national Social Prescribing Link Worker Survey 2025." Please make necessary edits.

***Thank you for your time. We have responded to all the edits.

---

## [Decision Letter · Decision Letter 1]

17 Mar 2026

Patterns and predictors of outcome monitoring amongst link workers: learnings from the national Social Prescribing Link Worker Survey 2025

PONE-D-25-67268R1

Dear Dr. Fancourt,

We’re pleased to inform you that your manuscript has been judged scientifically suitable for publication and will be formally accepted for publication once it meets all outstanding technical requirements.

Kind regards,

Souparno Mitra, M.D.

Academic Editor

PLOS One

Additional Editor Comments (optional):

Reviewers' comments:

Reviewer's Responses to Questions

**Comments to the Author**

Reviewer #1: All comments have been addressed

Reviewer #2: All comments have been addressed

2. Is the manuscript technically sound, and do the data support the conclusions?

Reviewer #1: Yes

Reviewer #2: Partly

3. Has the statistical analysis been performed appropriately and rigorously?

Reviewer #1: Yes

Reviewer #2: I Don't Know

4. Have the authors made all data underlying the findings in their manuscript fully available?

Reviewer #1: Yes

Reviewer #2: Yes

5. Is the manuscript presented in an intelligible fashion and written in standard English?

Reviewer #1: Yes

Reviewer #2: Yes

Reviewer #1: Thank you for responding to all my comments.

I believe that this manuscript is now ready for publication.

Reviewer #2: Thanks for making suggested edits on article " Patterns and predictors of outcome monitoring amongst link workers: learnings from the national Social Prescribing Link Worker Survey 2025."

.

Reviewer #1: **Yes:** Hendrik NapieralaHendrik NapieralaHendrik NapieralaHendrik Napierala

Reviewer #2: **Yes:** Arun George PrasadArun George PrasadArun George PrasadArun George Prasad

---

## [Editor Report · Acceptance letter]

PONE-D-25-67268R1

PLOS One

Dear Dr. Fancourt,

I'm pleased to inform you that your manuscript has been deemed suitable for publication in PLOS One. Congratulations! Your manuscript is now being handed over to our production team.

Kind regards,

on behalf of

Dr. Souparno Mitra

Academic Editor

PLOS One